# Immunomodulatory Effects of Sodium Hyaluronate Health Drink on Immunosuppressed Mice

**DOI:** 10.3390/foods13060842

**Published:** 2024-03-09

**Authors:** Xiaozhen Peng, Xiaoqiao Yao, Ya Liu, Bangzhu Peng

**Affiliations:** 1School of Public Health & Laboratory Medicine, Hunan University of Medicine, Huaihua 418000, China; peng112112@163.com; 2College of Food Science and Technology, Huazhong Agricultural University, Wuhan 430070, China; y1592434697@163.com (X.Y.); liuyaqf@163.com (Y.L.); 3Shenzhen Institute of Nutrition and Health, Huazhong Agricultural University, Wuhan 430070, China; 4Shenzhen Branch, Guangdong Laboratory for Lingnan Modern Agriculture, Genome Analysis Laboratory of the Ministry of Agriculture, Agricultural Genomics Institute at Shenzhen, Chinese Academy of Agricultural Sciences, Shenzhen 518120, China

**Keywords:** immunomodulatory, sodium hyaluronate, health drink, intestinal flora, immune deficiency

## Abstract

This study aimed to explore the immunomodulatory effects and mechanism of a sodium hyaluronate health drink in immunosuppressed mice. The results showed that the sodium hyaluronate health drink could improve thymus atrophy, repair spleen cell damage, promote the release of IL-2, IL-6 and TNF-α in serum, restore immune deficiency, and enhance immune function. In addition, 16s rRNA sequencing results of intestinal flora showed that different doses of health drink had different effects on the intestinal flora of mice. The low-dose group of mice showed a significant up-regulation of the abundance of *Lactobacillus* and promoted the formation of a new genus *Akkermansia*, while the medium- and high-dose group up-regulated the abundance of *Lactobacillus* and *norank-f-Muribaculaceae*, and stimulated the production of the new genus *Alistipes*. Sodium hyaluronate health drink may enhance the immune function of mice by changing the composition and abundance of intestinal flora, which provided a theoretical basis for the subsequent product development.

## 1. Introduction

The management of immunosuppression utilized in malignancies and severe manifestations of various auto-immune diseases requires highly nuanced care. The prompt identification and treatment of these disorders are crucial in reducing both mortality and morbidity rates. Cyclophosphamide is a common immunosuppressive drug used in clinical practice and trials. However, its therapeutic application is limited by its side effects. The administration of cyclophosphamide has been demonstrated to have deleterious effects on normal tissue DNA, resulting in the destruction of healthy immune cells and the inhibition of the proliferation and differentiation of macrophages, B and T lymphocytes [1]. This can ultimately lead to the suppression of both humoral and cellular immune responses, as well as immune dysfunction and intestinal injury. Specifically, cyclophosphamide has been shown to disrupt the intestinal mucosal barrier, compromise the integrity of the epithelium and neighbouring cell–cell junctions, and impair the absorption of nutrients in the intestine [2,3,4,5].

Confronted with these limitations, there is no targeted treatment option, so the intake of health foods that enhance immunity has become the best choice for these people to fight against the decline of immunity. Studies have proved that many polysaccharides have significant immunomodulatory effects, which is the main reason why many foods are currently reported to enhance immune function. Xie et al. demonstrated that *Anoectochilus formosanus* polysaccharide has the potential to mitigate cyclophosphamide-induced immunosuppression, enhance their thymus index and body weight, stimulate cytokine production, and promote intestinal morphology repair [6]. Wang et al., found that Cicadas polysaccharide extracted with ultrasound-assisted extraction could regulate host immune response [7]. Liu found that the polysaccharide of the Opuntia stem could effectively improve the metabolism level of lysine synthesis and decomposition in immunodeficient mice and improve the health level of immunodeficient mice [8].

However, polysaccharides cannot be directly digested and absorbed in the intestine, and are mostly regulated by intestinal flora. Alterations in the polysaccharide sources within the intestinal microbiota frequently play a role in immunomodulatory processes. Studies have found that longan polysaccharides can enhance host immune function under stress conditions by regulating intestinal flora and intestinal metabolites [9]. Ginger polysaccharide has been shown to enhance cytokines IL-2, IL-4, TNF-α, and immunoglobulin Ig-G secretion in the serum of mice, thereby ameliorating immune deficiency. Additionally, ginger polysaccharide has been found to promote the proliferation of beneficial bacterial taxa, such as *Muribaculaceae, Bacteroidaceae*, and *Lactobacillaceae*, while concurrently decreasing the relative abundance of detrimental bacterial populations like *Rikenellaceae* and *Lachnospiraceae* [10]. Cordyceps polysaccharide has been shown to enhance the diversity of the intestinal microbial community, modulate the overall structure of intestinal microflora, induce the secretion of cytokines and the synthesis of transcription factors, and effectively exhibit protective effects against intestinal mucosal immunosuppression and microbial ecological imbalance in mice [11]. Curcuma polysaccharide demonstrated the ability to enhance thymus and spleen indices, elevated whole blood immune cell and lymphoid count indices, and promoted the secretion of serum immunoglobulin IgG. Moreover, it exhibited potential in repairing intestinal immune damage and mitigating intestinal inflammation [12]. Obviously, bioactive polysaccharides supported the growth of beneficial bacterial populations in the gut, bolstered the host immune system, and the gut flora derived from polysaccharides interacts with immune regulation.

Sodium hyaluronate, as a highly physiologically active polysaccharide, was originally isolated from the vitreous humour of bovine eyes [13]. Its known physiological functions include antioxidant [14], cosmetic and moisturizing [15], immune and anti-inflammatory [16,17], anti-tumour [18], and weight loss [19]. Several studies have shown that sodium hyaluronate is involved in regulating the immune response of the body through various channels [20]. This provides new insights into the physiological and pathological mechanisms by which sodium hyaluronate is involved in immune and inflammatory responses. Sodium hyaluronate is often used as a multifunctional antigen carrier for the targeted delivery of nanomedicines, indirectly involved in human immunomodulation [21]. The immunomodulatory effect of sodium hyaluronate is mostly related to the production of stimulated related cytokines, and whether its intake affects the composition of the body’s intestinal flora is unclear, nor is it clear whether it participates in the body’s immune regulation through the intestinal flora. Therefore, it is of great significance to elucidate the immunomodulatory function of the sodium hyaluronate health drink from the perspective of intestinal flora and explore its immune mechanism.

## 2. Materials and Methods

### 2.1. Materials and Reagents

Sodium hyaluronate as a food-grade powder, with a molecular weight of 120 kDa, was purchased from Henan Sanhua Biotechnology Co., Ltd. (Ruzhou, China). Citric acid was purchased from Shandong Weifang Ensign Industrial Co., Ltd. (Weifang, China). White sugar was purchased from Shandong Yahui Sugar Industry Co., Ltd. (HeZhe-, China). Vitamin C was purchased from CSPC Group Weisheng Pharmaceutical Co., Ltd. (Shijiazhuang, China). NaHCO_3_ was purchased from Hebei Zhongnong Bailiang E-Commerce Co., Ltd. (Baoding, China). Paraformaldehyde (4%) was purchased from Wuhan Seville Biotechnology Co., Ltd. (Wuhan, China). Mianyang red blood cells (2%) were purchased from Shanghai Yudore Biotechnology Co., Ltd. (Shanghai, China). Drabkins solution was purchased from Haibiao Technology Co., Ltd. (Xiamen, China). Indian ink was purchased from Soraibao Biotechnology Co., Ltd. (Beijing, China).

### 2.2. Preparation of Sodium Hyaluronate Health Drink

In this experiment, a health drink based on sodium hyaluronate has been developed as a sample for in vivo experiments in mice. The health drink was prepared as follows: sodium hyaluronate powder was mixed with drinking water and stirred in a magnetic stirrer for 2 h until the sodium hyaluronate was completely dissolved to obtain sodium hyaluronate solution. The formula of the sodium hyaluronate health drink is: 0.03% sodium hyaluronate, 0.03% NaHCO_3_, 4.5% sugar, 0.1% citric acid, 0.05% vitamin C, and 0.03% edible flavour.

### 2.3. Immunological Animal Experimental Design

Sixty SPF grade BALB/C male mice (20.0 ± 2.0 g) were purchased from the Animal Centre of Huazhong Agricultural University (Wuhan, China, Certificate number: SCXK (Hubei) 2020-0084). The mice were kept in a clean environment with a stable ambient temperature of 22 ± 2 °C and maintained on a 12 h day/night cycle, and immunological tests were performed after 7 d of acclimatization [6]. The whole process of mouse testing is in strict compliance with the national regulations on test animals. The concentrated health drink was produced according to the manufacturing process of this health drink, in which the sample for the low-dose group was a 2-fold concentrated sodium hyaluronate health drink (sodium hyaluronate concentration of 0.6 g/L); the sample for the medium-dose group was a 4-fold concentrated sodium hyaluronate health drink (sodium hyaluronate concentration of 1.2 g/L); and the sample for the high-dose group was an 8-fold concentrated sodium hyaluronate health drink (sodium hyaluronate concentration of 2.4 g/L). The gavage volume of each mouse was based on its daily body weight (gavage volume (mL) = body weight (g)/100). The mice were randomly and equally divided into 6 groups: normal control (NC), model control (MC), positive control (PC), low dose (LD), medium dose (MD) and high dose (HD). The low-, medium-, and high-dose groups were gavaged with the corresponding concentrations of sodium hyaluronate health drink, the positive control group was gavaged with 100 mg/kg/d of lentinan, and the rest of the groups were gavaged with normal saline. The drug was administered using gavage daily for 35 days (5 weeks). All groups except the normal control group were treated for immune depression using intraperitoneal injection of 80 mg/kg of cyclophosphamide solution according to body weight three days before the first week of gavage administration and two days before the rest of the week, while the normal control group was injected with an equal amount of normal saline. The body weights of the mice were recorded before the start of the test (day 0) and each day after gavage, respectively. The mice were fasted for 12 h after the final dose, and the eyes were removed to obtain blood, and then they were executed. Our study was approved by the Animal Ethics Committee of Huazhong Agricultural University (ID Number: HZAUMO-2021-0163), China.

### 2.4. Immunological Test Method

After the cervical vertebrae were dislocated and sacrificed, the thymus and spleen were removed as soon as possible, washed with normal saline, dried with filter paper, and weighed, respectively. The thymus index and spleen index of mice were calculated as follows: I = W_1_/W_2_(1)
where I represents organ index; W_1_ represents the mass of the organ; and W_2_ is the weight of the mouse.

The spleens of mice were divided into two halves and placed in prenumbered 4% paraformaldehyde fixation solution. After standing and fixing for 24 h, sections were embedded and stained with HE, and then photographed under an inverted microscope.

The humoral immune function of the mice was demonstrated using the measurement of serum haemolysin. Four days before the end of the test, each mouse was immunised using an intraperitoneal injection of 0.2 mL of 2% pressed sheep red blood cells (SRBC) suspension. After 4 days, the eyeballs were removed for blood collection, and the serum was collected using centrifugation at 1000–2000× rpm for 10 min and stored at −80 °C. The serum was diluted 200 times with normal saline in a test tube; 1 mL of the diluted serum was aspirated into a new test tube, and then 0.5 mL of 10% (*v*/*v*) SRBC and 0.5 mL of guinea pig serum complement (diluted 1:8 in saline) were added in turn and mixed well. A control tube without serum was set up (replaced with saline). After mixing, the tubes were placed in a constant temperature water bath at 37 °C for 25 min and then removed to an ice bath for 5 min to terminate the reaction. After the ice bath, the solution in the tubes was transferred to a centrifuge tube and centrifuged at 2000 rpm for 10 min. In total, 1 mL of the supernatant was taken into the tube, and 3 mL of Drabkins solution was added and mixed, while 0.25 mL of 10% (*v*/*v*) SRBC was added to 4 mL of Drabkins solutionand mixed as half of the lysate. The tubes were mixed well and left to stand for 10 min, and the absorbance of each tube was measured at 540 nm using the control tube as a blank. The HC_50_ of the samples was calculated according to the following equation: HC_50_ = As/Ah(2)
where HC_50_ represents half hemolysis value; As represents sample tube absorbance value; and Ah represents half lysis tube absorbance value.

The function of monocyte-macrophages was determined by means of a carbon contour test in mice. Mice were injected with Indian ink diluted five times with saline in the tail vein, and the ink was injected and immediately timed. At 2 min and 10 min after ink injection, 20 mL of blood was taken from each of the internal adjacent venous plexus and immediately added to a pre-prepared tube containing 2 mL of 0.1% Na_2_CO_3_ solution. The absorbance was measured with a spectrophotometer at 600 nm using Na_2_CO_3_ solution as a blank control. The mice were executed and the liver and spleen were taken, blotted with filter paper to remove blood from the surface of the organs and weighed separately. The phagocytosis index α was calculated according to following equation: K = (lgOD_1_ − lgOD_2_)/(t_2_ − t_1_)(3)
α = body weight ÷ (liver weight + spleen weight) × 3√K(4)
where OD_1_ and OD_2_ represent the absorbance at 2 min and 10 min, respectively, and t_1_ and t_2_ represent 2 min and 10 min, respectively.

After eyeball removal, blood was collected and left for 1 h, centrifuged at 3500 rpm for 10 min, and the upper layer of serum was divided and stored in liquid nitrogen. The levels of immune factors IL-2, IL-6, and TNF-α were measured according to the kit. 

### 2.5. Animal Experimental Design for Intestinal Flora

The mice were randomly divided into four groups of 10 mice each, namely the normal control group (NC), low-dose group (LD), medium-dose group (MD), and high-dose group (HD). The low-, medium-, and high-dose groups were gavaged with the corresponding concentrations of sodium hyaluronate health drink, while the rest of the groups were gavaged with normal saline. The drug was administered by gavage for 28 days (4 weeks). Body weight was recorded once a week. The faeces of each mouse were taken in six parallel portions and packed in sterile tubes, snap frozen in liquid nitrogen, and then quickly transferred to −80 °C for storage. The samples of each group were selected as NC (NC1, NC2, NC4, NC5, NC7, NC8), LD (LD1, LD2, LD5, LD6, LD7, LD9), MD (MD1, MD2, MD3, MD4, MD5, MD6), HD (HD3, HD4, HD5, HD6, HD7, HD8). Following the conclusion of the experimental period, all animals underwent a 12 h fasting period prior to euthanasia via an intraperitoneal injection of ketamine hydrochloride (15 mg/kg/i.p.) and xylazine (10 mg/kg/i.p.).

### 2.6. 16S rRNA Sequencing Method

A total of four groups of six samples each were obtained using 16S rRNA sequencing. The 16S rRNA Illumina sequencing technology was applied to analyse the effect of this health drink on faecal flora. The DNA was first extracted from the faeces, the quality and concentration of the DNA was tested using NanoDrop2000, the integrity of the DNA was tested using a 1% agarose gel, and the DNA obtained was used as a template for PCR amplification. 

Primer sequences: forward primer: 5′-ACTCCTACGGGAGGCAGCAG-3′; reverse primer: 5′-GGACTACHVGCCTWTCTAAT-3′. PCR was performed on the V3-V4 region. The amplification procedure was as follows: pre-denaturation at 95 °C for 3 min, 27 cycles (denaturation at 95 °C for 30 s, annealing at 55 °C for 30 s, extension at 72 °C for 45 s), followed by stable extension at 72 °C for 10 min, and finally storage at 10 °C (PCR instrument: ABI GeneAmp^®^ Model 9700, (Thermo Fisher, Waltham, MA, USA). After passing the test, the library was sequenced on an Illumina Hiseq2500 platform (Illumina, San Diego, CA, USA).

### 2.7. Statistical Analysis

SPSS 17.0 (IBM, Armonk, NY, USA) statistical software was used for the analysis, and one-way ANOVA was chosen to compare statistical differences between groups. All data were expressed as mean ± standard deviation (SD), and *p* < 0.05 was considered as a significant difference.

## 3. Results 

### 3.1. Body Weight and Organ Index of Mice 

The body weights of the mice were recorded in Figure 1A, and the results showed that the body weight of each group of mice decreased significantly after intraperitoneal injection of cyclophosphamide, indicating that cyclophosphamide would cause the body weight of mice to plummet. After administration, the body weight of mice in each group increased slowly, which was not as good as that of the NC group. And, except for the HD group, the other groups did not increase their body weight before modelling, which may be related to the influence of cyclophosphamide on the growth and development of organisms (Figure 1B). In addition, there was no death or abnormal phenomenon in mice during the experiment, indicating that this health drink was harmless to the health of mice.

The effect of the sodium hyaluronate health drink on the immune organs of mice is shown in Table 1. There was no significant difference in the spleen index of mice (*p* > 0.05), and the thymus index of mice moulded with cyclophosphamide was significantly lower than that of the NC group (*p* < 0.05), indicating that cyclophosphamide caused atrophy of the thymus. The thymus indexes of mice in the MD and HD groups were significantly higher (*p* < 0.05) compared to those in the MC group, indicating that thymus atrophy in mice in the MD and HD groups improved after 5 weeks of oral gavage of different doses of this health drink.

### 3.2. Spleen Histomorphology of Mice

To further visualise the structural changes in the spleen of immunocompromised mice, the spleen was stained with HE and photographed in sections, and the results are shown in Figure 2A. The two main functional areas of the spleen were called the white and red marrow, and the white and red marrow of the spleen of the NC mice were very well defined, and the splenocytes were well distributed and closely arranged. In contrast, the spleen of the MC group had a blurred border between the red and white marrow, which was almost impossible to distinguish, and there was obvious cellular oedema, similar to ulceration, and a large number of splenocytes were oedematous and distributed in a disorganized manner. Compared with the MC group, the proportion of white marrow in the spleen of mice in the PC group was significantly increased, the cellular edema was relieved, and the arrangement of spleen cells was more compact and orderly. The health drink showed a similar trend to the PC group in all dose groups, but the proportion of white marrow was not as high as that of the PC group. 

In order to further observe the distribution and arrangement of splenocytes in the spleen of mice, the morphology of splenocytes in each group of mice was further analysed under high magnification (Figure 2B). The splenocytes of the NC group were arranged in a tight and orderly manner, showing a swirling arrangement. In the MC group, the splenocytes were loosely arranged irregularly, and the number of cells decreased. The arrangements of splenocytes in the various doses of health drink groups were more orderly and compact than those of the model group, and the swirling shape was gradually obvious, approaching the splenocyte state of normal mice, but the arrangement of splenocytes in the PC group was not obvious. The number of splenocytes in the PC group and the health drink group all increased significantly compared with that in the MC group. It showed that this health drink could repair the spleen cell damage caused by cyclophosphamide.

### 3.3. Immunology Experiments

The haemolytic capacity is judged using colourimetric determination, and the results are usually expressed using the hemolytic value (HC_50_), with higher HC_50_ values indicating greater haemolytic capacity and immunity. The effect of the sodium hyaluronate health drink on the serum haemolysis of mice is shown in Figure 3A. The HC_50_ of mice in the MC group was significantly lower than that of the NC group (*p* < 0.01), which showed that the moulding was successful and the immune damage caused by cyclophosphamide to the lymphocytes of mice was great. In the MD and HD groups, the serum haemolysin production capacity was significantly enhanced compared to that of the MC group (*p* < 0.05), with the MD group having the most significant effect in enhancing serum haemolysin production capacity. It could be seen that this health drink helped to improve the humoral immune function of mice.

A delayed allergic reaction is also known as a “delayed hypersensitivity reaction”, which occurs 24 h after exposure to an antigen, hence the term “delayed hypersensitivity reaction”. In this test, mice were stimulated with sheep red blood cells to induce an allergic reaction. The extent of the reaction was mainly reflected in the swelling of the foot topography, with higher swelling indicating a more severe allergic reaction and stronger immunity. The effect of this health drink on the delayed allergic reaction in mice is shown in Figure 3B. After injection of cyclophosphamide mice stimulated with sheep red blood cells, the degree of foot swelling was significantly lower than that in the NC group, which was due to the cellular immunodeficiency caused by cyclophosphamide. The foot swelling degree of mice after the gastric lavage health drink increased significantly compared with the MC group (*p* < 0.01), indicating that long-term intake of this health drink could repair the immune deficiency caused by cyclophosphamide, among which the tissue swelling level of mice in the LD group was higher than that in the NC group, which showed that oral administration of this health drink not only had the effect of repairing low immunity but had the function of enhancing immunity.

The phagocytosis speed of the phagocytes is directly proportional to the concentration of carbon particles. Considering the different sizes of mouse organs, the phagocytosis index is usually used to express the phagocytic activity per unit of tissue weight, and the level of phagocytosis index reflects the strength of the phagocytic ability of macrophages in mouse organs. The effect of the sodium hyaluronate health drink on the phagocytic index of macrophages in mice is shown in Figure 3C. The phagocytic index of mice in the MC group decreased compared with that in the NC group (*p* > 0.05), which showed that cyclophosphamide had an impairing effect on the phagocytic ability of macrophages in mice. The macrophage phagocytosis of immunocompromised mice was restored after oral administration of this health drink, but there was no significant difference between the dose groups and the MC group (*p* > 0.05), indicating that this health drink did not have a significant effect on enhancing the macrophage phagocytosis of immunocompromised mice.

### 3.4. Serum Levels of Immune Factors

IL-2, IL-6, and TNF-α are all extremely important immune factors in immune regulation and are involved in the body’s immune regulation in a variety of ways [19]. The level of TNF-α immune factor is closely related to tumours, diabetes, and especially autoimmune diseases [22,23,24]. As can be seen from Figure 3D–F, the content of IL-2 in the serum of mice modelled with cyclophosphamide was significantly lower than that of mice in the NC group (*p* > 0.05); the serum contents of IL-6 and TNF-α in the MC group were significantly lower than those in the NC group (*p* < 0.01). The contents of IL-2, IL-6, and TNF-α in the serum of mice in the MD group and HD group of the sodium hyaluronate health drink after 5 weeks of gavage were significantly enhanced (*p* < 0.05 and *p* < 0.01). It might be seen that the sodium hyaluronate health drink could stimulate the release of immune factors in mice, and among which the IL-6 and TNF-α content in immunocompromised mice gradually returned to normal levels after gavage of the sodium hyaluronate health drink, which indicated that long-term gavage of this sodium hyaluronate health drink could effectively restore the immunodeficiency caused by cyclophosphamide.

### 3.5. Dilution Curve and Alpha and Beta Diversity

The Rarefaction curve reflects the microbial diversity of each sample at different sequencing volumes. The trend of the curve can reflect the homogeneity, richness, or diversity of a species in a sample, as well as the reasonableness of the amount of sequencing data. The horizontal coordinate of the curve is generally the amount of data randomly selected, while the vertical coordinate corresponds to the Alpha diversity index. The curve flattens out as the sample size increases, indicating that the sample is homogeneous and the amount of data is sufficient. The dilution curve of this experiment was plotted with Sobs index as the vertical coordinate and the amount of randomly selected data as the horizontal coordinate, as shown in Figure 4. The level of Sobs index in the faeces of each group of mice increased rapidly and then levelled off as the number of randomly selected samples increased. The Shannon curve was also plotted with the Shannon index as the vertical coordinate; the Shannon index levelled off and then did not increase significantly as the number of randomly selected samples increased, which indicated that the amount of experimental data was reasonable and large enough to be of analytical significance.

Alpha diversity analysis reflects the overall richness and diversity of the microbial community, with Chao, Ace, Shannon, and Simpson as the main measures. Table 2 showed the variation of Alpha diversity in the intestinal flora of mice, where the Shannon and Simpson indices were used to reflect the diversity of the community in the sample, while the Chao and Ace indices were used to reflect the abundance of the community. As shown in Table 2, the abundance of intestinal flora in mice after gavage of this health drink was smaller than that of the NC group, but the diversity of flora increased. The Ace and Chao indices in the LD group were significantly lower than those in the NC group (*p* < 0.01 and *p* < 0.05), the Ace and Chao indices in the MD group were not significantly different from those in the NC group (*p* > 0.05), and the Ace and Chao indices in the HD group were significantly lower than those in the NC group (*p* < 0.01). This reflected that gavage of both low and high doses of the health drink resulted in a reduction in the abundance of mouse flora. In contrast, the Simpson index of mice in the LD group was significantly increased (*p* < 0.01) compared with the NC group, the Shannon index and Simpson index of mice in the MD group were significantly decreased (*p* < 0.01) and significantly increased (*p* < 0.05), respectively, and the Shannon index and Simpson index of mice in the HD group were very significantly different from those in the NC group (*p* < 0.01), which showed a decrease in the Shannon index and an increase in the Simpson index, of which the Shannon index decreased less than the increase in Simpson, which indicated that the colony species diversity of mice increased after gastric lavage. Richness and diversity are often regarded as important indicators of the homeostasis of intestinal flora, but not desiring higher richness and diversity, as the more ideal situation is a dynamically balanced homeostatic environment. The coverage of each group in Table 2 was higher than 0.999, indicating that all species in the sample were detected, further verifying that the number for sequencing was sufficient. The difference between groups was tested using a statistical *t* test, as shown in Figure 5A, and there was a significant difference between the index of each dose group and the normal group (*p* < 0.05). 

Similarly, a Beta cluster analysis was performed on the samples to obtain Figure 5C, which showed that the samples in each dose group were significantly different from the NC group, with the samples in the MD group being more similar to those in the HD group.

### 3.6. Species Composition Analysis

The Venn plot mainly reflects the number of common and unique species (such as OTU) in multiple groups of samples, and it mainly reflects the similarity of species composition (such as OTU) in different environmental samples through overlap. The number of common and unique OTUS between the normal group and each dose group of health drinks was analysed according to the study requirements, and the Venn chart was plotted. The numbers in the overlapping part of the graph represent the number of species common to the overlapping groups and the numbers in the non-overlapping part represent the number of species unique to the corresponding groups. The Venn diagrams for the four groups of samples are shown in Figure 5B, where the number of OTUS common to each group was 206, the number of OUT unique to the NC group was 21, the number of OUT unique to the LD group was 9, the number of OUT unique to the MD group was 19, and the number of OUT unique to the HD group was 10. It could be concluded that some new microbial species would be produced in the intestinal tract of mice after intaking the sodium hyaluronate health drink, and the richness of new bacterial genera was very important for the balance of intestinal flora.

To further investigate the microbial community composition and species richness of the four groups of samples, the species richness of each group of samples was calculated at the phylum, family, and genus levels and plotted as a histogram, as shown in Figure 6. From Figure 6A, it can be seen that at the phylum classification level, the intestinal flora of mice consisted mainly of *Bacteroidetes* (Phylum *Bacteroidetes*) and *Firmicutes* (Phylum Thick-walled Bacteria), which accounted for more than 90% of the total flora. The LD group of health drinks had an increase in *Firmicutes* compared with the NC group, and the number of *Bacteroidetes* decreased compared with the NC group. The abundance of *Firmicutes* in the MD group decreased compared with that in the NC group, corresponding to an increase in *Bacteroidetes*. The HD group showed a similar trend to the MD group. Studies have shown that there is a correlation between the number of *Firmicutes* in the gut and the body’s obesity status, with an increase in the body’s diet and an increase in body fat content causing a subsequent increase in the number of *Firmicutes*, which is why *Firmicutes* are called “Fat Fungus” [22]. Ley first showed that obesity was associated with intestinal flora and that obese mice had more *Firmicutes* than lean mice [23]. At the same time, Million disputed this view, arguing that it was mostly gut microbes at the species level that correlate more strongly with obesity status in humans [24]. There are several hypotheses as to the specific mechanisms by which gut flora contribute to obesity, but there are no definitive findings. In short, changes in gut flora are involved in the development of obesity. The differences between the remaining phylum groups were small and not discussed.

At the family level, the abundance of *muribaculaceae* in the LD group was lower than that in the NC group, while the abundance of *muribaculaceae* in the MD and HD groups of the health drink was higher than that in the NC group. The abundance of *Lactobacillaceae* in each dose group was increased compared with that in the NC group, and the increase was more obvious in the LD group. The abundance of *Lachnospiraceae* and *Bacteroidaceae* in the MD group was lower than that in the NC group.

At the genus level, compared with the NC group, the abundance of *norank-f-Muribaculaceae* and *Bacteroides* was down-regulated in the LD group, while the abundance of *Lactobacillus* was significantly up-regulated. The MD group up-regulated the abundance of *norank-f-Muribaculaceae* and *Lactobacillus*, and down-regulated the abundance of *Bacteroides*. The HD group up-regulated the abundances of *norank-f-Muribaculaceae* and *Lactobacillus*, and down-regulated the abundances of *Bacteroides* and *Staphylococcus*.

### 3.7. Species Variation Analysis

The multi-species differences of intestinal microbiota in each group of mice are shown in Figure 6D. As shown in Figure 6D, the abundance of the genus *norank-f-Muribaculaceae* was highly significantly different between the groups (*p* < 0.01). Compared to the NC group, the abundance of this genus was significantly up-regulated in the MD group and HD group, while it was decreased in the LD group. The abundance of *Lactobacillus* in the LD group was significantly up-regulated, and that in the MD and HD group was less up-regulated. The abundance of *Bacteroides* decreased slightly in the LD group compared to the NC group, decreased significantly in the HD group, and decreased very significantly in the MD group; the abundance of *Turicibacter* (involved in fermentation metabolism, and its main metabolites have muscle regulating and anti-fatigue effects) showed significant upward adjustment in the MD group. *Alistipes* is a butyric acid-producing, specialized anaerobic bacterium. The Butler study suggested that its abundance may be inversely associated with obesity [25], and its abundance was significantly up-regulated in the MD group, almost exclusively in mice in the MD and HD groups. The abundance of *Akkermansia* was significantly up-regulated in the LD group, and as a mucin-degradable bacterium, it was inversely associated with obesity, diabetes, inflammation, and metabolic disorders [26], and the up-regulation of the abundance of the genus was beneficial to health. The abundance of the remaining genera is small and will not be discussed here.

LEfSe analysis reflects genetic, metabolic, and taxonomic profiles and is often used to distinguish between multiple groups of organisms. The results of the LDA discriminant plots of the intestinal flora structure of different groups of mice are shown in Figure 6E. It could be seen that the bacteria in the NC group that had a greater influence on the change of flora were *Clostridia*, *Bacteroidaceae,* and *Bacteroides*. In the LD group, the most influential groups were *Firmicutes*, *Bacilli*, *Lactobacillales,* and their species. In the MD group, the groups that had a greater impact on the change in flora were *Muribaculaceae*, *Bacteroidales*, *Bacteroidota,* and *Citrobacter*. In the HD group, *Paludicola* had a greater effect on the change in flora. It might be seen that the oral administration of different doses of this health drink could affect the structural changes of the organism’s flora, and affect different species of intestinal flora.

## 4. Discussion

In this study, we found that sodium hyaluronate supplements had a modulating effect on immune function in immunocompromised mice. In addition, we found that sodium hyaluronate drinks had the effect of regulating the intestinal flora of mice, and the effects of different doses on the flora of mice were different.

In the past few decades, a large number of biologically active polysaccharides have been discovered and put into practical use, and immunomodulatory activity is the physiological activity of most polysaccharides. The difference is that different polysaccharides have different structures and are absorbed by the body in different forms, and their mechanism of action in humans are different [27]. Therefore, a large number of scholars have investigated the immunomodulatory mechanism of polysaccharides through animal experiments. The spleen index and thymus index are indicative of the development of immune organs and are commonly utilized as key parameters for assessing overall immune system functionality [28]. In our study, we found that sodium hyaluronate health drinks have immune-enhancing effects on the body, mainly in terms of repairing apparent damage to organs (Table 1, Figure 2). The immune system is a defence system composed of humoral immunity, cell immunity, and non-specific immunity, etc. [29]. The findings of the hemolysin test indicated a significant increase in serum hemolysin production in the MD and HD groups (Figure 3A). Moreover, the impact of the sodium hyaluronate health drinks on the delayed allergic response in mice was demonstrated in Figure 3B. In addition, the phagocytic index of mice in the MC group decreased compared with that in the NC group, which indicated that the health drinks could enhance the phagocytosis of macrophages (Figure 3C). These results indicated that the oral administration of this health drink not only demonstrated efficacy in restoring low immunity levels but also exhibited the capacity to enhance overall immune function. The immune system produces and secretes cytokines that play a role in regulating the immune system to maintain a healthy immune system [10]. To further explore sodium hyaluronate health drinks’ immunoregulatory mechanism, the findings indicated that the serum levels of IL-2, IL-6, and TNF-α in mice treated with sodium hyaluronate beverage exhibited alterations compared to those in the control group (Figure 3D–F), suggesting an enhancement in immune response. These results were consistent with the immune-enhancing effects of many polysaccharides. 

Numerous studies have shown that most large polysaccharides are not absorbed in the stomach and small intestine; they are often partially or completely hydrolysed and absorbed by the body through the combined action of intestinal microorganisms [30]. It is commonly believed that polysaccharides could be converted into short-chain fatty acids and carbon monoxide via the intestinal flora, which are further involved in the body’s energy metabolism and immune response [31], and the absorption mechanism of sodium hyaluronate, as a mucopolysaccharide, in the human body has long been a hot issue for research.

Indeed, the small and large intestines of healthy humans are home to trillions of bacteria, viruses, archaea, and fungi [32]. It is widely accepted that the foetus remains largely sterile during development, the only sterile stage of human growth and development [33]. *Bifidobacterium* and *Lactobacillus* are the main dominant microorganisms colonising the gut in the short term after birth, and these lactic acid metabolising bacteria are mainly of maternal origin, gradually increasing in species with age [34], gradually building on the initial microbial base to form a large gut microbial community [35]. The microbiota significantly contribute to the maintenance of immune homeostasis [10]. This is mainly due to the fact that the gut has a specific immune system and has multiple layers of protection to limit the ability of resident microbes to enter the systemic circulation and thus the systemic immune system [36]. Mariko’s study found that after a single oral hyaluronic acid treatment for 8 h in 7–8 week old male SD rats, hyaluronic acid was present in large amounts in the intestinal contents of the rats, implying that sodium hyaluronate is potentially involved in the immunomodulatory processes of the body through the intestinal flora [37]. In this study, the intestinal microbiome of mice was profoundly affected by sodium hyaluronate health drinks. Administration of both low and high doses of the health drink led to a decrease in the abundance of mouse microbiota. Conversely, the diversity of colony species in mice increased following gastric lavage (Table 2). The low-dose group of mice exhibited a noteworthy increase in the prevalence of *Lactobacillus* and facilitated the emergence of a novel taxonomic group, *Akkermansia*, whereas the medium- and high-dose groups demonstrated an increase in the prevalence of *Lactobacillus* and *norank-f-Muribaculaceae*, and induced the production of the new taxonomic group *Alistipes* (Figure 6). A large amount of the literature has documented a positive correlation between the relative abundance of *Lactobacillus* and *Akkermansia* and immunity [10]. *Lactobacillus,* as the most common probiotic, has been used in various beverages for a long time to regulate the balance of the body’s intestinal flora, and the proliferation of *Lactobacillus* can promote intestinal absorption, protect the gastric mucosa, and play a positive regulatory role in maintaining human intestinal immunity. In the case of *Alistipes*, a new genus of *Alistipes* emerged in the MD and HD group after long-term gavage of this health drink (Figure 6D), which is considered to be rarely involved in human diseases [38] and may be negatively associated with human obesity [39], mainly due to the positive correlation between *Alistipes* and serum ANGPTL4 and adropin. Serum ANGPTL4, a secreted glycoprotein, has been shown to inhibit lipoprotein lipase activity, while the adropin protein has been shown to play an important role in regulating energy metabolism and maintaining insulin sensitivity, and has been found to be significantly increased in high-fat-fed mice [25]. The rise in both protein levels is closely related to the metabolism of fat, which explains the possible mechanism of the weight loss effect of sodium hyaluronate on the body from the perspective of microorganisms, and the excessive accumulation of fat is related to cardiovascular and cerebrovascular diseases, which in turn affects the human immune system, but how it is possible to affect the immune system from the perspective of protein is not clear. As shown in Figure 6D, a new genus of *Akkermansia* appeared in the mice of the LD group. By colonizing with *Akkermansia muciniphila* in germ-free mice, Derrien confirmed that the genus increased the expression of genes involved in the immune response and determining cell fate, and found that it regulated pathways involved in immune tolerance in the symbiotic microbiota, and participated in the establishment of a homeostatic environment for basal metabolism [40]. In short, the intricate relationship between the sodium hyaluronate health drink, bacterial species, microbial metabolic pathways, and immune indices sheds light on the potential mechanism of host immune response to polysaccharides at the level of intestinal microbiota.

## 5. Conclusions

In this study, we investigated the immunomodulatory activity of a sodium hyaluronate health drink in immunocompromised mice through immunological tests, and elaborated the effect of the health drink on intestinal microflora of mice from the perspective of intestinal microflora. It was found that long-term intake of the sodium hyaluronate health drink could enhance the immune function of mice. Compared with the model group, the phagocytosis ability of spleen cells of mice was enhanced after drinking the health drink, and the serum hemolysin production ability of mice in the MD and HD groups was significantly higher than that in the model group, increasing by about 10%. In the sequencing of intestinal flora, it was found that the abundance of *Muribaculaceae* and *Lactobacillaceae* in each dose group was significantly increased, and the proportion of *Lactobacillaceae* in intestinal flora was also significantly increased, which confirmed that gavage of the sodium hyaluronate health drink could promote the growth of intestinal probiotics in mice. The intake of different doses of the sodium hyaluronate health drink could stimulate the production of different probiotics in the intestinal tract of mice, thus regulating intestinal balance and participating in the immune regulation of mice.

## Figures and Tables

**Figure 1 foods-13-00842-f001:**
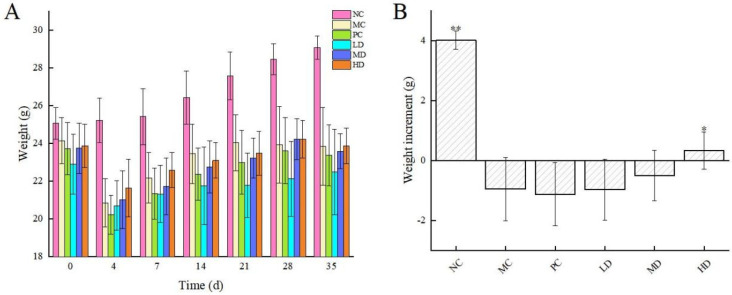
(**A**) Body weight in mice; (**B**) body weight gain of mice. The values are presented as mean *±* SD, *n* = 10. Compared to initial weight, * indicates a significant difference (*p* < 0.05), and ** indicates a highly significant difference (*p* < 0.01).

**Figure 2 foods-13-00842-f002:**
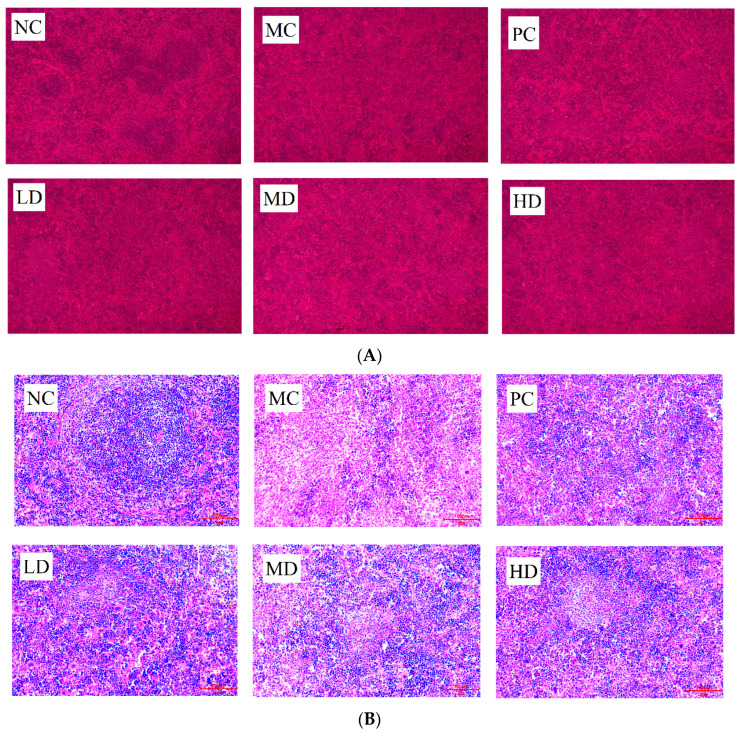
Section of spleen tissue (hematoxylin and eosin staining, *n* = 10): (**A**) (100×); (**B**) (200×).

**Figure 3 foods-13-00842-f003:**
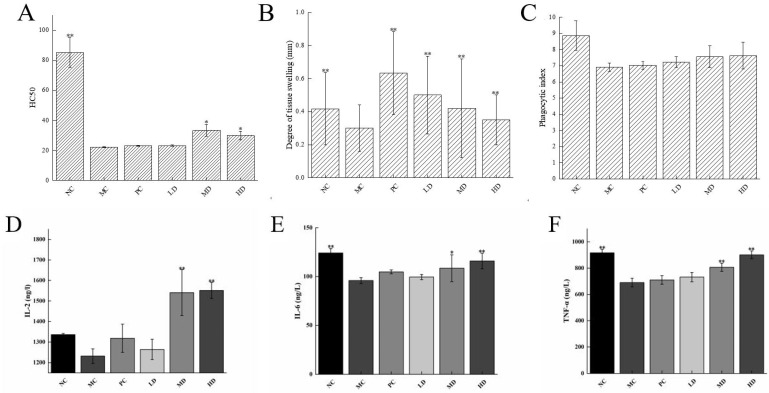
(**A**) Ability to produce serum hemolysin; (**B**) delayed-type hypersensitivity; (**C**) carbon clearance phagocytic index; serum levels of immune factor (**D**) IL-2, (**E**) IL-6, (**F**) TNF-α in mice. The values are presented as mean *±* SD, *n* = 10. Compared to NC group, * indicates a significant difference (*p* < 0.05), and ** indicates a highly significant difference (*p* < 0.01).

**Figure 4 foods-13-00842-f004:**
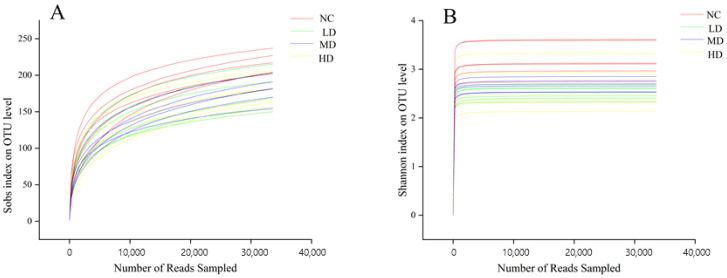
(**A**) Sobs index dilution curve; (**B**) Shannon index dilution curve. *n* = 6.

**Figure 5 foods-13-00842-f005:**
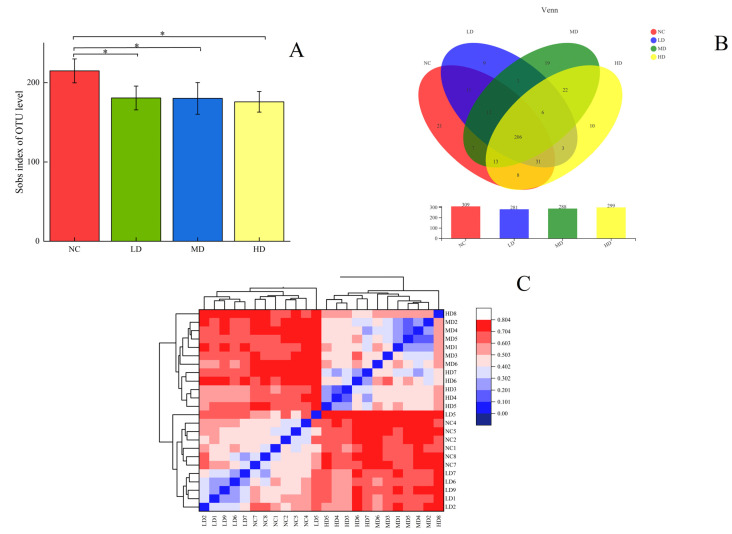
(**A**) Plot of difference between groups at OTU level; (**B**) Venn graph; (**C**) Beta cluster analysis diagram. The values are presented as mean *±* SD, *n* = 6. Compared to NC group, * indicates a significant difference (*p* < 0.05). One-way ANOVA was chosen to compare statistical differences between groups.

**Figure 6 foods-13-00842-f006:**
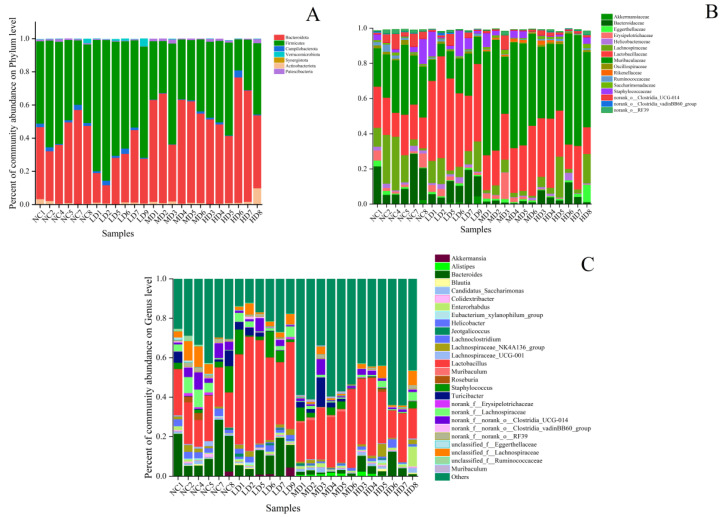
The relative abundances of the major bacteria at (**A**) phylum level, (**B**) family level, (**C**) genus level based 16S rRNA sequences. (**D**) Multispecies difference test bar chart; (**E**) LDA discriminant tree. *n* = 6. One-way ANOVA was chosen to compare statistical differences between groups. Compared to NC group, * indicates a significant difference (*p* < 0.05), and ** indicates a highly significant difference (*p* < 0.01), *** indicates a highly significant difference (*p* < 0.001).

**Table 1 foods-13-00842-t001:** Effect of health drink on spleen index and thymus index in mice.

Groups	Spleen Index (mg/g)	Thymus Index (mg/g)
NC	4.01 ± 0.03	2.54 ± 0.01 *
MC	3.85 ± 0.04	1.67 ± 0.02
PC	4.39 ± 0.06	1.93 ± 0.01
LD	4.08 ± 0.02	1.91 ± 0.01
MD	4.40 ± 0.04	2.07 ± 0.03 *
HD	4.25 ± 0.03	2.06 ± 0.02 *

Note: The values are presented as mean *±* SD, *n* = 10. Compared with MC group, * means *p* < 0.05, showing significant difference.

**Table 2 foods-13-00842-t002:** Alpha diversity index analysis table.

Sample	Richness		Diversity		Coverage
	Ace	Chao	Shannon	Simpson	
NC	252.59 ± 17.55	255.59 ± 23.59	3.19 ± 0.31	0.09 ± 0.02	0.9992
LD	213.25 ± 20.09 **	217.33 ± 24.75 *	2.52 ± 0.13 **	0.21 ± 0.05 **	0.9993
MD	227.87 ± 25.23	230.46 ± 30.10	2.67 ± 0.11 **	0.14 ± 0.02 *	0.9990
HD	207.86 ± 18.13 **	206.81 ± 17.24 **	2.60 ± 0.41 **	0.16 ± 0.05 **	0.9992

Note: The values are presented as mean *±* SD, *n* = 6. Compared to NC group, * indicates a significant difference (*p* < 0.05), and ** indicates a highly significant difference (*p* < 0.01).

## Data Availability

The original contributions presented in the study are included in the article, further inquiries can be directed to the corresponding author.

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
