# Peer review of "Immunomodulatory Effects of Sodium Hyaluronate Health Drink on Immunosuppressed Mice"

_foods, 2024, doi:10.3390/foods13060842_

Round 1

Reviewer 1 Report

Comments and Suggestions for Authors

In the paper entitled “Immunomodulatory Effects of Sodium Hyaluronate Health Drink on Immunosuppressed Mice” the authors describe the effect of sodium hyaluronate health drink on the modulation of the immune response and the regeneration of the intestinal microbiota in cyclophosphamide-induced immunosuppression in mice. The work is conceptually well designed, through the analysis of immune parameters and the analysis of changes in the diversity and functionality of the intestinal microbiota. However, in my view, there are some omissions and ambiguities. ...Some parts of the article require more serious changes.

 1. Abstract

 Please make a change in the sentence:

“This study was aimed to explore the immunomodulatory effect and mechanism of sodium 12 hyaluronate health drink in immunomodulatory mice.

The sentence should read:

This study aimed to explore the immunomodulatory effect and mechanism of sodium 12 hyaluronate health drink in immunosuppressed mice.

Introduction

The introductory part is insufficiently explained. I believe that the introductory part of the article lacks a part about the most common reasons that cause immunosuppression, as well as what is the basis of immunosuppression induced by an alkylating compound such as cyclophosphamide, especially on the immune system and intestinal epithelial cells, changes in the intestinal microbiome and the importance of preserving intestinal homeostasis. It should also be emphasized the effect and importance of the use of detoxifying and protective agents with the purpose of enhancing the efficacy and reducing the toxicity of cyclophosphamide and other toxins and drugs that change the intestinal ecosystem leading to gut disbiosis. It should be emphasized a bidirectional interplay between gut microbiota and many organs within the human body such as the intestines, the lungs, the brain, and the skin.

Materials and Methods

1.  According to the guidelines for acclimatization of newly admitted laboratory animals, rodent species should have a minimum acclimation period of 3 days (72 hours) prior to use.I think that longer periods for acclimation, conditioning, and/or training of animals may be required for testing immune function, given that stress greatly reduces the immune response and increases the level of stress hormones such as cortisol/corticosterone,  but also affects other bodily functions.

​2. I believe that the plan of the experiment should be shown through the Figure; the beginning of the induction of immunosuppression of mice with cyclophosphamide and the period of treatment with sodium hyaluronate health drink so that the method and time of treatment were clearer, as well as the period of taking samples for analysis.... It is difficult to follow this way.

3.  Furthermore, I believe that the treatment period with sodium hyaluronate health drink is quite long considering the metabolism and lifespan of a mouse compared to a human.How do you explain the long time of administration of hyaluronate health drink for 5 weeks via gastric cannula. Isn't that too stressful for the animals and considering the human-mouse lifespan ratio is 1 year for humans versus 9 days or in the adult phase, 2.6 mice days are equivalent to one human years?

4. It should be emphasized how to choose the dose of cyclophosphamide and sodium hyaluronate health drink. Explain how the doses used in mice correspond to the doses used in humans?    What is the basis for the chosen doses? I strongly suggest the authors to explain whether the concentrations of sodium hyaluronate health drink used in the mice model experiments are achievable doses for human therapy? Please indicate the appropriate dose for humans, where it should except for body surface area involved, and many other parameters such as several parameters of biology, including oxygen utilization, caloric expenditure, basal metabolism, blood volume, circulating plasma proteins, and renal function. Please explain in “Material and Methods” and “Discussion” sections.

4. There are methodological errors in the paper. In the sentence: After 4 days, the eyeballs were removed for blood collection, and the serum was collected by centrifugation for 1 h and stored at -80℃.

The procedure is wrong, 10 minutes is enough to obtain serum from coagulated blood. The clot should be removed by centrifugation at 1,000-2,000 x g for 10 minutes in a refrigerated centrifuge.

5. In the Materials and methods lack the method of anesthetizing the animals.

6. Explain the choice of positive control and the dose used (100 mg/kg/d of mushroom polysaccharide).

7. Figures are insufficiently described and unclear without reading the text. The Figures should contain all the important information to be understandable without reading the entire paper. The method of treating the mice, the period of treatment and animal sacrifice, the number of mice per group, the method of analysis and statistical processing were not specified.

8. In Figure 2, there is no description of the obtained changes, methods of staining and analysis of samples, number od samples…

9. The analysis of the intestinal microbiota is well done, but additional information should be added in the description of the Figure about the method of treatment, analysis, number of samples, etc.

Discussion
Furthermore, the Discussion also requires major revisions and a better review of its own results and their mutual relationship, as well as a link to the results from previous literature on this topic. In the entire Discussion, it seems as if only one sentence gives an overview of the results obtained. I think that the obtained results should be clearly highlighted, and connected through the possible effects of sodium hyaluronate health drink and its effect on immune cell function and
and changes in intestinal microbiota. There is a lack of a link to existing literature and a clear emphasis on additional mechanisms of sodium hyaluronate health drink and its role in preventing immunosupression. In the Discussion, there is no reference anywhere to the obtained results (through the specified Table or Figure) and a clear presentation of the immunomodulating and regenerative role through the intestinal microbiota.  In the Discussion, it should be clarified how the resulting changes in the intestinal microbiota affect the immune system and the regeneration of the gut... What are the key mechanisms of their mutual relationship? Perhaps it would be good to point out the possible limitations of the conducted research.

References

References are not written according to the journal's instructions. Please read the journal's instructions for authors carefully.References must be numbered in order of appearance in the text.

Conclusions

It is OK.

Second, minor problems, there are some mistakes about editorial handling, e.g. in vitro, in vivo and p value must be written italic, SI units should be used (μL or mL). Uniform the way of writing abbreviations, p value (P or p), name of drugs, figure (Fig. or Figure) and units. Please separate the SI unit from the number. The sentence should not begin with a number. Abbreviations must be explained the first time they are used, both in the Abstract and again in the main text. Abbreviations should be after the Abstract and in the middle of the Introduction section. There are many typographical and grammatical errors. Recheck them carefully!

Decision:

I agree that it is a good idea but the scientific parameters are poorly incorporated into the text and insufficiently explained.I think that the authors of this paper do not meet the criteria of the Foods journal. In my opinion, the paper should be completely restructured, supplemented and resubmitted.

Author Response

Dear Editors and Reviewers:

Thank you for your decision and constructive comments on my manuscript. I hope that the changes I’ve made resolve all you concerns about the article. Revision notes, point-to-point, are given as follows:

  1. Please make a change in the sentence: “This study was aimed to explore the immunomodulatory effect and mechanism of sodium 12 hyaluronate health drink in immunomodulatory mice.”

Answer: Sorry for this mistake, this sentence was revised by “This study aimed to explore the immunomodulatory effect and mechanism of sodium 12 hyaluronate health drink in immunosuppressed mice.”

  1. The introductory part is insufficiently explained. I believe that the introductory part of the article lacks a part about the most common reasons that cause immunosuppression, as well as what is the basis of immunosuppression induced by an alkylating compound such as cyclophosphamide, especially on the immune system and intestinal epithelial cells, changes in the intestinal microbiome and the importance of preserving intestinal homeostasis. It should also be emphasized the effect and importance of the use of detoxifying and protective agents with the purpose of enhancing the efficacy and reducing the toxicity of cyclophosphamide and other toxins and drugs that change the intestinal ecosystem leading to gut disbiosis. It should be emphasized a bidirectional interplay between gut microbiota and many organs within the human body such as the intestines, the lungs, the brain, and the skin.

Answer: We have improved our Introduction as your suggestion.

  1. According to the guidelines for acclimatization of newly admitted laboratory animals, rodent species should have a minimum acclimation period of 3 days (72 hours) prior to use. I think that longer periods for acclimation, conditioning, and/or training of animals may be required for testing immune function, given that stress greatly reduces the immune response and increases the level of stress hormones such as cortisol/corticosterone, but also affects other bodily functions.

Answer: Sorry for this mistake. The mice were provided with a standard diet for a period of 7 days in order to achieve metabolic stabilization before being allocated into distinct groups. Xie et al. also administered a standard diet to mice for a period of seven days, subsequently segregating the animals into two distinct groups (Protective Effect of Anoectochilus formosanus Polysaccharide against Cyclophosphamide-Induced Immunosuppression in BALB/c Mice).

  1. I believe that the plan of the experiment should be shown through the Figure; the beginning of the induction of immunosuppression of mice with cyclophosphamide and the period of treatment with sodium hyaluronate health drink so that the method and time of treatment were clearer, as well as the period of taking samples for analysis.... It is difficult to follow this way.

Answer: We really appreciate your valuable comments and suggestion. We agree that the plan of the experiment Figure would be useful to understand the details of interaction and enhancement. However, completing the Figure is challenging due to the constraints of limited time.

  1. Furthermore, I believe that the treatment period with sodium hyaluronate health drink is quite long considering the metabolism and lifespan of a mouse compared to a human.How do you explain the long time of administration of hyaluronate health drink for 5 weeks via gastric cannula. Isn't that too stressful for the animals and considering the human-mouse lifespan ratio is 1 year for humans versus 9 days or in the adult phase, 2.6 mice days are equivalent to one human years?

Answer: Thank you very much for the valuable advice. A 5-week period of administration of hyaluronic acid health drinks via gastric cannula is reasonable. Wang et al. conducted a study in which Sanziguben polysaccharides were administered to a mouse model of diabetic nephropathy for a duration of 8 weeks. (Sanziguben polysaccharides improve diabetic nephropathy in mice by regulating gut microbiota to inhibit the TLR4/NF-κB/NLRP3 signalling pathway).

  1. It should be emphasized how to choose the dose of cyclophosphamide and sodium hyaluronate health drink. Explain how the doses used in mice correspond to the doses used in humans? What is the basis for the chosen doses? I strongly suggest the authors to explain whether the concentrations of sodium hyaluronate health drink used in the mice model experiments are achievable doses for human therapy? Please indicate the appropriate dose for humans, where it should except for body surface area involved, and many other parameters such as several parameters of biology, including oxygen utilization, caloric expenditure, basal metabolism, blood volume, circulating plasma proteins, and renal function. Please explain in “Material and Methods” and “Discussion” sections.

Answer: Thank you very much for the valuable advice. The dosage of sodium hyaluronate health drink was calculated according to the body weight of human. We will suggest the scope of sodium hyaluronate health drink. Further experiments are required to determine if the concentration of sodium hyaluronate in the health drink used in the mouse model experiment is equivalent to the therapeutic dose achievable in humans, which is the focus of our upcoming research.

  1. There are methodological errors in the paper. In the sentence: After 4 days, the eyeballs were removed for blood collection, and the serum was collected by centrifugation for 1 h and stored at -80℃.

Answer: Sorry for this mistake, this sentence was revised by “After 4 days, the eyeballs were removed for blood collection, and the serum was collected by centrifugation 1,000-2,000 x g for 10 minutes and stored at -80℃. ”

  1. In the Materials and methods lack the method of anesthetizing the animals.

Answer: Thank you for your suggestion. We have added the method of anesthetizing the animals in the Materials and methods.

  1. Explain the choice of positive control and the dose used (100 mg/kg/d of mushroom polysaccharide).

Answer: Mushroom polysaccharides play an important role in regulating animal immune function through stimulating natural killer cells involving neutrophils and macrophage dependent immune system responses, in addition to modifying receptors such as those of dectin-1, toll-like receptor-2, scavengers and lactosylceramides. The dosage of mushroom polysaccharide at 100 mg/kg/d was determined through a review of relevant literature and subsequent experimental validation. (Influence of Mushroom Polysaccharide, Nano-Copper, Copper Loaded Chitosan, and Lysozyme on Intestinal Barrier and Immunity of LPS-mediated Yellow-Feathered Chickens, etc.)

  1. Figures are insufficiently described and unclear without reading the text. The Figures should contain all the important information to be understandable without reading the entire paper. The method of treating the mice, the period of treatment and animal sacrifice, the number of mice per group, the method of analysis and statistical processing were not specified.

Answer: Thank you for your suggestion. We have improved our Figures as your suggestion.

  1. In Figure 2, there is no description of the obtained changes, methods of staining and analysis of samples, number od sample…

Answer: Thank you for your suggestion. We have revised Figure 2 as your suggestion. The description of the obtained changes: The two main functional areas of the spleen were called the white and red marrow, and the white and red marrow of the spleen of the NC mice were very well defined, and the splenocytes were well distributed and closely arranged. In contrast, the spleen of the MC group had a blurred border between the red and white marrow, which was almost impossible to distinguish, and there was obvious cellular oedema, similar to ulceration, and a large number of splenocytes were oedematous and distributed in a disorganized manner. Compared with the MC group, the proportion of white marrow in spleen of mice in PC group was significantly increased, the cellular edema was relieved, and the arrangement of spleen cells was more compact and orderly. The health drink showed a similar trend to the PC group in all dose groups, but the proportion of white marrow was not as high as that of the PC group (Figure 2A). The splenocytes of the NC group were arranged in a tight and orderly manner, showing a swirling arrangement. In the MC group, the splenocytes were loosely arranged irregularly, and the number of cells decreased. The arrangement of splenocytes in the various doses of health drink group was more orderly and compact than that of the model group, and the swirling shape was gradually obvious, approaching the splenocyte state of normal mice, but the arrangement of splenocytes in PC group was not obvious. The number of splenocytes in the PC group and the health drink group all increased significantly compared with that in the MC group. It showed that this health drink could repair the spleen cell damage caused by cyclophosphamide (Figure 2B).

  1. The analysis of the intestinal microbiota is well done, but additional information should be added in the description of the Figure about the method of treatment, analysis, number of samples, etc.

Answer: Thank you very much for the valuable advice. We have revised Figures as your suggestion.

  1. Furthermore, the Discussion also requires major revisions and a better review of its own results and their mutual relationship, as well as a link to the results from previous literature on this topic. In the entire Discussion, it seems as if only one sentence gives an overview of the results obtained. I think that the obtained results should be clearly highlighted, and connected through the possible effects of sodium hyaluronate health drink and its effect on immune cell function and and changes in intestinal microbiota. There is a lack of a link to existing literature and a clear emphasis on additional mechanisms of sodium hyaluronate health drink and its role in preventing immunosupression. In the Discussion, there is no reference anywhere to the obtained results (through the specified Table or Figure) and a clear presentation of the immunomodulating and regenerative role through the intestinal microbiota. In the Discussion, it should be clarified how the resulting changes in the intestinal microbiota affect the immune system and the regeneration of the gut... What are the key mechanisms of their mutual relationship? Perhaps it would be good to point out the possible limitations of the conducted research.

Answer: Thank you very much for the valuable advice. We have revised Discussion as your suggestion.

  1. References are not written according to the journal's instructions. Please read the journal's instructions for authors carefully. References must be numbered in order of appearance in the text.

Answer: Thank you very much for the valuable advice. We have revised References as your suggestion.

  1. Second, minor problems, there are some mistakes about editorial handling, e.g. in vitro, in vivo and p value must be written italic, SI units should be used (μL or mL). Uniform the way of writing abbreviations, p value (P or p), name of drugs, figure (Fig. or Figure) and units. Please separate the SI unit from the number. The sentence should not begin with a number. Abbreviations must be explained the first time they are used, both in the Abstract and again in the main text. Abbreviations should be after the Abstract and in the middle of the Introduction section. There are many typographical and grammatical errors. Recheck them carefully!

Answer: Thank you very much for the valuable advice. We have revised References as your suggestion.

Reviewer 2 Report

Comments and Suggestions for Authors

The article submitted to us is very interesting to read, but certain recommendations should be made to improve the quality of the manuscript:

*      The problematic needs to be improved because as presented it does not highlight the interest of the study. In our view, the authors need to identify the causes of immune system failure. Consequently, the researchers focused on discovering compounds that positively or negatively modulate the biological response of immune cells and improve the host's ability to resist infection. As such, several classes of these compounds, such as polysaccharides, have all been characterized as molecules with potent effects on the host immune system.

*      Line 39 needs to be revisited, as it does not address the main issue or is poorly stated. We think that the authors would do well to present the positive influence of polysaccharides on the intestinal microflora and their positive effect on the immune system. Line 42 to 54 need to be summarized.

*      According to which mixing plan was the dietary solution made? Provide the mixing plan. A mixing plan was necessary in order to obtain the ideal dietary solution. Failing this, the authors should carry out acute and sub-acute toxicity tests beforehand to ensure that the product is safe. These tests are necessary to improve the quality of the article.

*      Please provide the approval number of the ethics committee.

*      Authors must add the bibliographical references that served as a basis for implementing the methodology. Otherwise, this work will have no scientific basis.

*      All tables and figures must be understandable in themselves. Consequently, abbreviations must be defined at the bottom of the figure or table.

*      In general, an effort should be made in the presentation of the results

*      The quality of figure 6 should be improved. Some graphs (A, B, C) should be reproduced in other formats or in table form to make the results easier to understand. It is not the beauty of the graphs that must prevail.

*      The authors say little about the Sodium Hyaluronate diet drink and what could justify its effectiveness

*      Transitions need to be improved

Author Response

  1. The problematic needs to be improved because as presented it does not highlight the interest of the study. In our view, the authors need to identify the causes of immune system failure. Consequently, the researchers focused on discovering compounds that positively or negatively modulate the biological response of immune cells and improve the host's ability to resist infection. As such, several classes of these compounds, such as polysaccharides, have all been characterized as molecules with potent effects on the host immune system.

Answer: Thank you very much for the valuable advice. Sodium hyaluronate is a highly physiologically active polysaccharide. We confirmed that gavage of sodium hyaluronate health drink could promote the growth of intestinal probiotics for regulate the intestinal balance, and participate in the immune regulation of mice.

  1. Line 39 needs to be revisited, as it does not address the main issue or is poorly stated. We think that the authors would do well to present the positive influence of polysaccharides on the intestinal microflora and their positive effect on the immune system. Line 42 to 54 need to be summarized.

Answer: Thank you very much for the valuable advice. We have revised the manuscript as your suggestion.

  1. According to which mixing plan was the dietary solution made? Provide the mixing plan. A mixing plan was necessary in order to obtain the ideal dietary solution. Failing this, the authors should carry out acute and sub-acute toxicity tests beforehand to ensure that the product is safe. These tests are necessary to improve the quality of the article.

Answer: Thank you very much for the valuable advice. The formula of the sodium hyaluronate health drink is: 0.03% sodium hyaluronate, 0.03% NaHCO3, 4.5% sugar, 0.1% citric acid, 0.05% vitamin C and 0.03% edible flavour.

  1. Please provide the approval number of the ethics committee.

Answer: The approval number of the ethics committee has been added in “Institutional Review Board Statement” of the revised Manuscruipt.

  1. Authors must add the bibliographical references that served as a basis for implementing the methodology. Otherwise, this work will have no scientific basis.

Answer: Thank you very much for the valuable advice. We have added the bibliographical references that served as a basis for implementing the methodology.

  1. All tables and figures must be understandable in themselves. Consequently, abbreviations must be defined at the bottom of the figure or table.

Answer: Thank you very much for the valuable advice. We have revised them as your suggestion.

  1. In general, an effort should be made in the presentation of the results

Answer: Thank you very much for the valuable advice. We have revised Results as your suggestion.

  1. The quality of figure 6 should be improved. Some graphs (A, B, C) should be reproduced in other formats or in table form to make the results easier to understand. It is not the beauty of the graphs that must prevail.

Answer: Thank you very much for the valuable advice. We have improved the quality of figure 6.

  1. The authors say little about the Sodium Hyaluronate diet drink and what could justify its effectiveness.

Answer: Thank you very much for the valuable advice. We have added the necessary information about Sodium Hyaluronate diet drink and how to justify its effectiveness.

  1. Transitions need to be improved.

Answer: Thank you for your advice, we have made meticulous modifications to this manuscript, therefore,the readers may understand our work more clearly.

Round 2

Reviewer 1 Report

Comments and Suggestions for Authors

In the manuscript “Immunomodulatory effects of sodium hyaluronate health drink on immunosuppressed mice” (Manuscript ID: foods-2855036R1), the authors did make a lot of requested corrections, but they ignored some of them, and I think that these data are necessary in scientific paper.

The first sentence in the abstract has not been changed. The sentence should read:

This study aimed to explore the immunomodulatory effect and mechanism of sodium 12 hyaluronate health drink in immunosuppressed mice.

In Materials and Methods in section 2.1. Materials and Reagents, the sentences starting with a number should be changed. According to spelling, sentences should not start with numbers or abbreviations.

Please change the following sentences:

4% paraformaldehyde was purchased from Wuhan Seville 204 Biotechnology Co., Ltd. 2% Mianyang red blood cells were purchased from Shanghai 205 Yudore Biotechnology Co., Ltd.

Please make the units in the same processes uniform. For example, the centrifugation process is shown somewhere in rpm, and somewhere as g.

I believe that it is necessary to state the dose of substances or drugs used, as well as the author according to which the stated doses were chosen when it comes to a similar experimental plan. Please provide the reference according to which mushroom polysaccharides were used as a positive control, but also for the dose of cyclophosphamide used.

Latin names should be written in italics, such as Lactobacillus or Akkermansia...

In the Discussion section, the authors mentioned and commented only on the results listed in Figures 1, 2 and 3. What about the other results, where were they commented? The results should be confirmed or rejected with already known mechanisms of literature. I believe that the explanation of one's own results should be clearly stated through a Table or Figure and supported by literature data or clarification of discrepancies with additional explanations of one's own results.

Author Response

  1. The first sentence in the abstract has not been changed.

Answer: Sorry for this mistake, this sentence was revised by “This study aimed to explore the immunomodulatory effect and mechanism of sodium hyaluronate health drink in immunosuppressed mice.”

  1. In Materials and Methods in section 2.1. Materials and Reagents, the sentences starting with a number should be changed. According to spelling, sentences should not start with numbers or abbreviations.

Please change the following sentences:

4% paraformaldehyde was purchased from Wuhan Seville 204 Biotechnology Co., Ltd. 2% Mianyang red blood cells were purchased from Shanghai 205 Yudore Biotechnology Co., Ltd.

Answer:Thank you for your suggestion. this sentence was revised by “ Paraformaldehyde (4%) was purchased from Wuhan Seville Biotechnology Co., Ltd. Mianyang red blood cells (2%)were purchased from Shanghai Yudore Biotechnology Co., Ltd.

  1. Please make the units in the same processes uniform. For example, the centrifugation process is shown somewhere in rpm, and somewhere as g.

Answer: Thank you for your suggestion. We have revised our manuscript as your suggestion.

  1. I believe that it is necessary to state the dose of substances or drugs used, as well as the author according to which the stated doses were chosen when it comes to a similar experimental plan. Please provide the reference according to which mushroom polysaccharides were used as a positive control, but also for the dose of cyclophosphamide used.

Answer: According to previous studies, a dosage of 100 mg/kg of mushroom polysaccharide were used as a positive control. Therefore, we opted to use 100 mg/kg of mushroom polysaccharide (lentinan) as our positive control in this study. (Reference: Polysaccharide of Alocasia cucullata Exerts Antitumor Effect by Regulating Bcl-2, Caspase-3 and ERK1/2 Expressions during Long-Time Administration)

Literature sources indicate that cyclophosphamide serves as an inducer in mouse immunosuppressive models, typically administered at a concentration of 80 mg/kg. (Reference:Reference: 1. Immunoenhancing Effects of Cyclina sinensis Pentadecapeptide through Modulation of Signaling Pathways in Mice with Cyclophosphamide-Induced Immunosuppression. 2. Immunomodulatory effect of oyster peptide on immunosuppressed mice).

  1. Latin names should be written in italics, such as Lactobacillus or Akkermansia...

Answer: Thank you for your suggestion. We have revised our manuscript as your suggestion.

  1. In the Discussion section, the authors mentioned and commented only on the results listed in Figures 1, 2 and 3. What about the other results, where were they commented? The results should be confirmed or rejected with already known mechanisms of literature. I believe that the explanation of one's own results should be clearly stated through a Table or Figure and supported by literature data or clarification of discrepancies with additional explanations of one's own results.

Answer: Thank you for your suggestion. We have revised the Discussion section as your suggestion.

Reviewer 2 Report

Comments and Suggestions for Authors

We find that the article manuscript has been considerably improved. The authors took our comments into account. At this stage, manuscript article can be accepted in present form.

Author Response

We find that the article manuscript has been considerably improved. The authors took our comments into account. At this stage, manuscript article can be accepted in present form.

Answer: We are very grateful for your kind appraisal. Thank you for your time and dedication in reviewing our manuscript. We are honored to have had the opportunity to receive feedback from someone with your level of expertise.

Once again, thank you very much for your comments and suggestions.